# Extracting the fingerprints of sequences of random rhythmic auditory stimuli from electrophysiological data

**Fernando A. Najman**[1], **Antonio Galves**[2†], **Marcela Svarc**[3], **Claudia D. Vargas**[4]*

**1** Instituto de Computação, Universidade Estadual de Campinas, Campinas, Brazil, **2** Instituto de Matemática e Estatística, Universidade de São Paulo, São Paulo, Brazil, **3** Departamento de Matemática y Ciencias, Universidad de San Andrés, Buenos Aires, Argentina, CONICET, Buenos Aires, Argentina, **4** Instituto de Biofísica Carlos Chagas Filho, Universidade Federal do Rio de Janeiro, Rio de Janeiro, Brazil

† Deceased.
\* cdvargas@biof.ufrj.br

**Data Availability Statement:** All relevant EEG data is accessible at: https://neuromat.numec.prp.usp. br/neuromatdb/EEGretrieving/. All source codes are

## Abstract

It has been classically conjectured that the brain assigns probabilistic models to sequences of stimuli. An important issue associated with this conjecture is the identification of the classes of models used by the brain to perform this task. We address this issue by using a new clustering procedure for sets of electroencephalographic (EEG) data recorded from participants exposed to a sequence of auditory stimuli generated by a stochastic chain. This clustering procedure indicates that the brain uses the recurrent occurrences of a regular auditory stimulus in order to build a model.

## Author summary

A classical conjecture is that the brain is constantly estimating regularities from sequences of events to be able to properly act upon the environment. We assume that, by doing statistics, the brain chooses a model from a class of possible models. Which class of models is used by the brain to encode sequences of events? We used an algorithm to generate a sequence of hand claps step by step reproducing a samba-like rhythm. These sequences were generated with stochasticity, where some auditory events were omitted with small probability. We retrieved the regularities of these random sequences of stimuli from EEG data recorded as the participants listened to the samba rhythm. To extract the information encoded in the EEG data we introduced a novel procedure for clustering sets of functional data by their relevant statistical features. The clusters obtained from the experimental data show that the strong beat of the rhythmic structure is used by the brain to encode the sequence. The strong beat has a remarkable property of separating the sequence into smaller independent blocks. This leads to a natural and economical explanation on how the brain organises the sequence in order to estimate the next event.

available at: https://github.com/fanajman/The-brain-uses-renewal-points-to-model-random-sequences-of-stimuli.

**Funding:** 2. This work is part of the activities of FAPESP Research, Innovation and Dissemination Center for Neuromathematics NeuroMat (grant # 2013/ 07699-0, São Paulo Research Foundation (FAPESP). FAN was supported by the Coordenação de Aperfeiçoamento de Pessoal de Nível Superior Foundation (CAPES) (88882.377124/2019-01) and FAPESP (2022/00784-0) grants. This work was also partially supported by the Conselho Nacional de Desenvolvimento Científico e Tecnológico (CNPq) fellowships (grants 314836/2021-7 to AG, 310397/2021-9 to CDV and 407092/2023-4 to CDV). CDV was also supported by Fundação Carlos Chagas Filho de Amparo à Pesquisa do Estado do Rio de Janeiro (FAPERJ, # CNE 202.785/2018 and # E- 26/010.002418/2019), and Financiadora de Estudos e Projetos (FINEP, # 18.569-8) grants. The funders had no role in study design, data collection and analysis, decision to publish, or preparation of the manuscript.

**Competing interests:** The authors have declared that no competing interests exist.

## Introduction

It has been proposed that the brain identifies statistical regularities in sequences of stimuli and organises these regularities to be able to classify the stimuli and to make predictions. How the brain proceeds to identify and to classify statistical regularities is still essentially an open question. Proposals on how the brain implements this issue have been suggested. For example, the brain may use Bayesian inference to estimate parameters relevant for the identification of statistical regularities [1–5]. Recently Duarte et al. [6] and Hernández et al. [7] proposed the use of probabilistic context tree models [8, 9] to represent the statistical regularities displayed by stochastic sequences of auditory stimuli. Employing this approach it was possible to extract, from EEG data, the regularities matching the algorithm used to generate the random sequence of stimuli. In [6, 7] the procedure to generate the sequence of auditory stimuli is defined by a context tree and an associated family of transition probabilities used to choose each next stimulus given the context associated to the sequence of past stimuli at each time step. A model selection procedure allowed Duarte et al. [6] and Hernández et al. [7] to retrieve from the EEG signals recorded at the prefrontal cortex the context tree governing the sequence of auditory stimuli.

In spite of the interest of all these works, each one of them assumes a particular class of models used by the brain to *organise* [10] the set of statistical regularities of sequences of stimuli. However, the identification of the class of models actually used by the brain remains an open question.

We propose to address this question by using a new clustering procedure for sets of EEG data collected while participants were exposed to sequences of auditory stimuli driven by a context tree. Clustering procedures have the property of retrieving any possible partition of sets of objects. Therefore, using a cluster-based scheme to model data should allow us to select a more parsimonious model of the EEG data in [7]. By identifying natural groupings in data, clustering analysis allows to identify patterns and relationships, with applications in pattern recognition, data compression, biological and medical research and social sciences [11]. There are many procedures to cluster data but there is no *one size fits all* strategy. The clustering technique depends on the nature and distribution of the data set. In neuroscience, clustering procedures have been applied solve challenges such as categorizing average EEG signals collected in different experimental conditions [12], or finding temporal segmentation in the EEG signals recorded from participants at rest [13]. However, none of them are suitable for clustering sets of functional data. In this paper we present a novel clustering procedure that addresses this problem. This clustering procedure has two stages. The first stage is at the individual level, where sets of segments of the EEG signal per participant are grouped using a new distance between the distribution of sets of functional data inspired by the projective method [14]. In the second stage, to summarize the individual results we employ a consensus clustering procedure. Consensus clustering has been successfully used in neuroscience to obtain accurate, stable and reliable grouping configurations (for example [15–17]). For more in depth review of consensus clustering see [18].

In our case, the sequence of stimuli used in the experiment can be characterized by the recurrent occurrence of a strong beat unit. As a matter of fact, in the context tree model employed in [7], the strong beat appears regularly and its appearance is sufficient for the brain to predict the next auditory stimulus. Thus, our conjecture is that the occurrence of the strong beat in the stimuli sequence yields a succinct partition of the underlying structure in EEG data. Our results show that regions of the pre-frontal cortex of the brain effectively use the occurrence of the strong beat to identify the structure embedded in the sequence of stimuli, with a more predominant role of the right hemisphere. Also, this new clustering procedure approach unveils hidden features of the data that could not be retrieved by a context tree [7].

## Materials and methods

### Ethics statement

All the participants gave their written consent in accordance with the relevant guidelines and regulations and approved by the Research Ethics Committee of the Deolindo Couto Institute of Neurology at the Federal University of Rio de Janeiro (Process Platform Brazil number 22047613.2.0000.5261).

### Experimental protocol

Nineteen participants were instructed to close their eyes, remain seated and listen carefully to a random sequence of auditory stimuli presented through headphones.

The auditory events used as stimuli were strong beats, weak beats, and silent units. The strong and weak beats were recordings of hand claps (spectral frequency range 0.2–15 KHz and maximum duration 200 ms each). The sound files were interrupted at 450 ms with a sharp cutoff. This was done using the software Audacity, version 2.0.5.0 (https://www.audacityteam.org/). The interstimulus interval between two consecutive sound units was always 450 ms.

The sequence of auditory events was chosen in a random fashion for each participant, described in the following section. EEG signals were recorded during the presentation of the sequences of auditory events. For more details on the experimental protocol see [7].

### Generating the stochastic sequence of auditory stimuli

Let the elements of the set {0, 1, 2} represent silences, weak beats, and strong beats, respectively. The sequence $X_n : n = 0, 1, \ldots, N$ can be generated symbol by symbol by using the following algorithm.

Let $\mathcal{V}$ be the set of all participants. For all $v \in \mathcal{V}$ we start with $X_0^v = 2$. For each $n = 0, \cdots, N - 1$, let $T_n^v$ be the largest $t$ such that $t \leq n$ and $X_t^v = 2$. Then:

- If $T_n^v$ is $n$ or $n - 2$, we set $X_{n+1}^v$ to 1 with probability 0.8, or 0 with probability 0.2.

- If $T_n^v = n - 1$, we set $X_{n+1}^v$ to 0.

- If $T_n^v = n - 3$, we set $X_{n+1}^v$ to 2.

This process generates stochastic sequences with a "samba-like" rhythm structure. A sample output is

$$\cdots 2\ 1\ 0\ 1\ 2\ 0\ 0\ 1\ 2\ 1\ 0\ 0\ 2\ 0\ 0\ 0\ 2 \cdots$$

Note that the label 2, that represents the strong beat stimulus, is a recurrent occurrence point; that is, for any $n$ and $k$ with $2 \leq n \leq n + k \leq N$,

$$\mathbb{P}(X_{n:n+k} = x_{n:n+k} | X_{n-1} = 2, X_{0:n-2} = x_{0:n-2}) = \mathbb{P}(X_{n:n+k} = x_{n:n+k} | X_{n-1} = 2);$$

where $X_{i:j}$ denotes the subsequence $(X_i, X_{i+1}, \ldots, X_j)$.

### Data acquisition and pre-processing

We used a Geodesic amplifier (Geodesic HidroCel GSN 128 EGI, Electrical Geodesic Inc.) coupled with high input impedance amplifier (200MΩ, Net Amps, Electrical Geodesics INC., Eugene, OR, USA). An analogical first order Butterworth band pass filter (0.3-50 Hz) was applied to the signal and the Cz electrode was used as the reference during data acquisition. The signal was acquired with recording frequency of 500 Hz.

In offline processing the data was re-referenced to the average using the EEGLAB package for MATLAB [19] and a fourth order Butterworth band pass filter (1-30 Hz) was applied to the signal.

In our analysis we used the electrodes $E = \{9, 10, 11, 18, 22, 74, 75, 82\}$, in the standard Geodesic numbering of 128 electrode sets [20]. For each electrode $e \in E$, each participant $v \in \mathcal{V}$, and each stimulus sequence index $n = 0, 1, \ldots, N$, we will denote by $Y_n^{e,v}$ the segment from that EEG signal starting 0.05 sec before the onset of that auditory stimulus and ending at 0.4 seconds after that onset. Formally assume that, for all $e \in \mathcal{E}$, $v \in \mathcal{V}$ and each $n \in \{1, \cdots, N\}$ we have that $Y_n^{e,v} \in L^2[-0.05, 0.4]$. The baseline of each EEG segment $Y_n^{e,v}$ was corrected by subtracting from it the average of the signal in the 50 ms immediately preceding the onset of $X_n$. For the sake of simplicity, we will not represent the baseline correction in the notation.

Furthermore, EEG signals recorded from single electrodes were combined in three subsets $E = \{\mathcal{E}^{\text{RPF}}, \mathcal{E}^{\text{LPF}}, \mathcal{E}^{\text{OCC}}\}$, namely:

$$\mathcal{E}^{\text{RPF}} = \{9, 10, 11\} \quad \mathcal{E}^{\text{LPF}} = \{11, 18, 22\} \quad \mathcal{E}^{\text{OCC}} = \{74, 75, 82\},$$

as corresponding to electrodes in the right prefrontal cortical region (RPF), left prefrontal cortical region (LPF) and occipital (OCC) region. The position in the array geodesic EEG cap is displayed in Fig 1.

The prefrontal electrodes were chosen given their proximity to the prefrontal cortex in the frontal lobe. The electrical activity at this region is known to change as a function of the presentation of an unlikely stimulus [21]. The source of the electrical activity observed in the prefrontal cortex employing improbable events has been shown to be lateralized [22]. For these reasons we opted to create two sets of electrodes, to the right and to the left of the frontal region, so as to guarantee that they were symmetrical with respect to the sagittal plane.

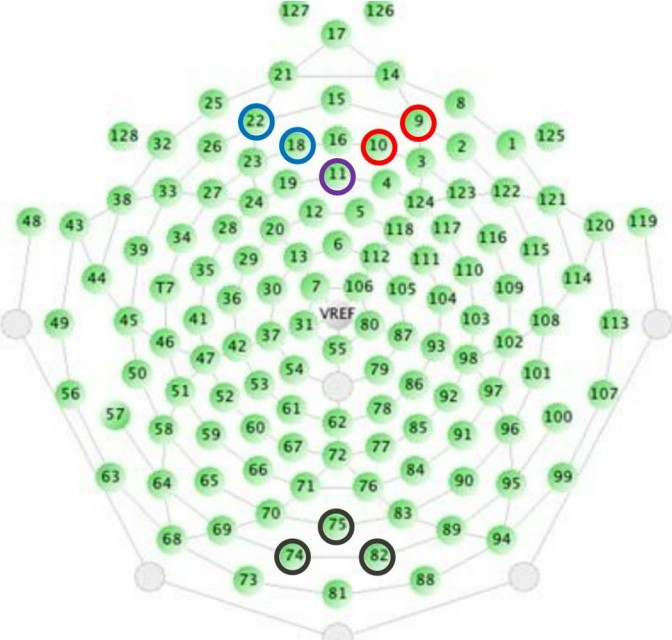

**Fig 1. Dense array geodesic EEG cap depicting in red the target electrodes over $\mathcal{E}^{\text{RPF}}$ (9 and 10), in blue those corresponding to $\mathcal{E}^{\text{LPF}}$ (18 and 22) and in purple the electrode (11) shared by both $\mathcal{E}^{\text{RPF}}$ and $\mathcal{E}^{\text{LPF}}$.** Control electrodes (74, 75 and 82) of the occipital region $\mathcal{E}^{\text{OCC}}$ are depicted in black.

Furthermore, to test the specific contribution of the common electrode 11 in the results and to explore whether the consensus partitions found in the prefrontal clusters show a lateralization component we used the sets $\mathcal{E}^{CNT} = \{10, 11, 18\}$, where *CNT* stands for central prefrontal, which is symmetrical to the sagittal plane in the frontal lobe, $\mathcal{E}^{LRPF} = \{9, 10\}$ and $\mathcal{E}^{LLPF} = \{18, 22\}$, where *LRPF* stands for lateralized right prefrontal and *LLPF* stands for lateralized left prefrontal. We added the occipital electrodes ($\mathcal{E}^{OCC}$) as a control since the occipital region is known to be mostly associated with visual processing [23].

For each subset $\mathcal{E} \in E$, each participant $v \in \mathcal{V}$, and each segment index $n$, we index by $\mathcal{E}$ the pointwise average $Y_n^{\mathcal{E}}$ of the signals $Y_n^e$ for all $e \in \mathcal{E}$; namely, for all $t \in [-0.05, 0.4]$, $Y_n^{\mathcal{E}}(t) = (\sum_{e \in \mathcal{E}} Y_n^e(t))/|\mathcal{E}|$, where $|\mathcal{E}|$ is the cardinality of the set $\mathcal{E}$.

## EEG data analysis

The goal was to retrieve from the EEG data the fingerprints of the putative model that the brain assigns to the sequence of auditory stimuli, see Fig 2. To achieve this task we introduce a new clustering procedure to group sets of EEG signals by their law.

**Clustering sets of EEG segments.** The first step of our analysis was to separately cluster sets of EEG segments of each participant $v \in \mathcal{V}$. To simplify notation, we will generally omit the index $v$ in $Y_n^{\mathcal{E},v}$ and $X_n^v$ in the remainder of this subsection.

Let $\mathcal{U}$ denote the set of all strings $X_n^{n+2}$ of three labels that may appear consecutively in any sequence $X_0, X_1, \ldots, X_N$. We denote by $P^*$ the partition of $\mathcal{U}$ determined by the position of the label 2 in the substring, or its absence. That is,

$$P^* = \{\{000, 001, 101, 100\}, \{200, 210\}, \{020, 021, 120, 121\}, \{002, 012\}\} .$$

This partition defines a four cluster structure. For each participant $v$, each set of electrodes $\mathcal{E}$,

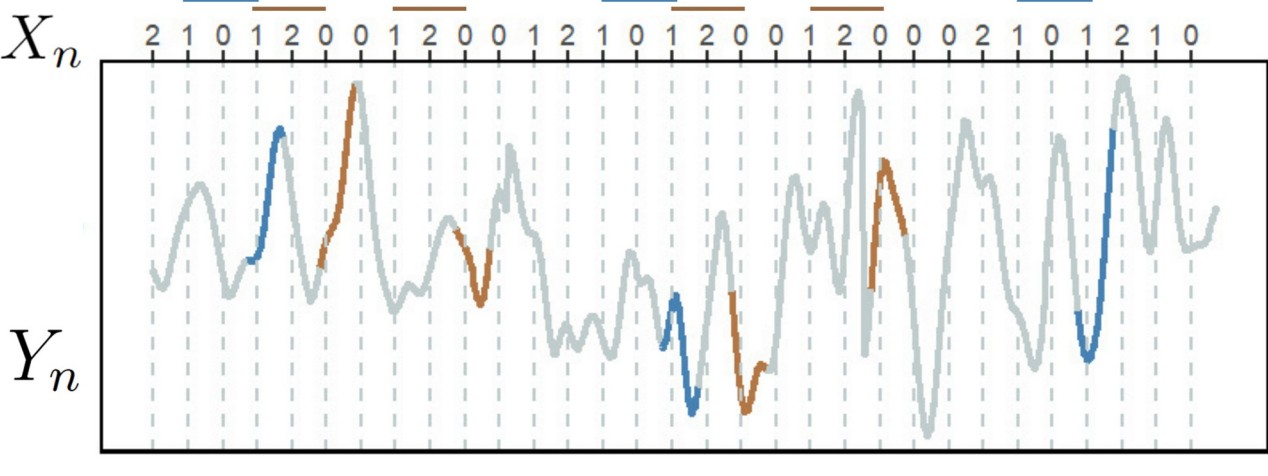

**Fig 2. Illustrative representation of the EEG signal $Y_n$ collected from one electrode while the participant was exposed to the sequence of auditory stimuli $X_n$.** We chose to represent two out of the twelve possible sequences of auditory stimuli. The blue EEG segments correspond to the string 101, while the brown ones, to the string 120. The goal was to retrieve from the EEG data the fingerprints of the putative model that the brain assigns to the sequence of auditory stimuli.

each string $u = (u_1, u_2, u_3) \in \mathcal{U}$, and for $n = 3, \ldots, N$ we denote by

$$\tilde{\mathcal{Y}}_N^{\mathcal{E},u} = \{Y_n^{\mathcal{E}}, n : 3, \ldots, N : X_{n-2}^n = u\}$$

the set of all EEG segments recorded during the presentation of stimulus $u_3$ whenever it followed stimuli $u_1$ and $u_2$. In this way, each EEG segment $Y_n^{\mathcal{E}}$ is associated with only a single string $u$. Fig 2 depicts this procedure.

As a preprocessing step, per string $u$, we ordered the EEG segments from $\tilde{\mathcal{Y}}_N^{\mathcal{E},u}$ using Fraiman-Muniz functional depth measure [24]. For each $t \in [-0.05, 0.4]$ this procedure assigns a pointwise inner-outer order of the EEG strings in the set $\tilde{\mathcal{Y}}_N^{\mathcal{E},u}$ using some depth measure in $\mathbb{R}$. To summarize this information, for each EEG string in $\tilde{\mathcal{Y}}_N^{\mathcal{E},u}$, the integral of the pointwise depths is computed over the interval $[-0.05, 04]$. Then EEG segments that are in the centre of the set attain high values while outlying EEG segments obtain lower values and are potentially flagged as outliers. This procedure allows removing segments contaminated by eye movement artifacts. We then denote by $\mathcal{Y}_N^{\mathcal{E},u}$ the set obtained by removing the ten percent most outlying segments from the set $\tilde{\mathcal{Y}}_N^{\mathcal{E},u}$.

We denote by $Q^{\mathcal{E},u}$ the distribution associated with the random EEG segments recorded in the subset $\mathcal{E}$ of electrodes during the presentation of the auditory stimuli indexed by $u_3$ occurring at the end of string $u = (u_1, u_2, u_3)$.

Given two strings $u$ and $u'$ and an arbitrary set of electrodes $\mathcal{E}$, with $u \neq u'$, we define a dissimilarity measure between the laws $Q^{\mathcal{E},u}$ and $Q^{\mathcal{E},u'}$. For simplicity sake we omit $\mathcal{E}$ in the notation unless otherwise indicated. Given a threshold $\delta \in (0, 1]$, the dissimilarity measure $\Delta^\delta$ between $Q^u$ and $Q^{u'}$ is defined by the formula

$$\Delta^\delta(Q^u, Q^{u'}) = \int_{L^2[T_0, T_1]} \mathbf{1}\{\|F_{Q,b} - F_{Q',b}\|_\infty > \delta\} dP(b),$$

where $F_{Q,b}$ is the marginal of $Q$ on the scalar product $y \in L^2([T_0, T_1]) \to \langle y, b \rangle$ where $b$ is drawn independently from the Gaussian measure $P$ and $\mathbf{1}\{A\}$ is the indicator function of condition $A$, then $\mathbf{1}\{A\} = 1$ if condition $A$ is satisfied and $\mathbf{1}\{A\} = 0$ otherwise. The integral is performed with respect to an independent Gaussian measure $P$. This follows from the results in [25]. Therefore the dissimilarity measure $\Delta^\delta$, based on the statistic corresponding to the Kolmogorov-Smirnov type goodness-of-fit test, allows to test whether two sets of square integrable functions have the same distribution. We consider the Brownian bridge as measure $P$, as in [7].

Since $\Delta^\delta(Q^u, Q^{u'})$ is not computable in practice, we estimate it using a finite sample of data employing the following procedure inspired by the projective method of Cuesta Albertos et al. [25]:

1. Let $B = (B(t): t \in [-0.05, 0.4])$ be a realisation of the Brownian bridge indexed by the time interval $[-0.05, 0.4]$, with $B(-0.05) = B(0.4) = 0$. The Brownian bridge $B$ is generated independently from the data set. For each participant $v$, each subset of electrodes $\mathcal{E} \subset E$ and each string $u$, we denote by $Z_n^{\mathcal{E},B,u}$ the real number defined as the inner product of $Y_n^{\mathcal{E},u} \in \mathcal{Y}_N^{\mathcal{E},u}$ and $B$. More precisely

$$Z_n^{\mathcal{E},B,u} = \int_{-0.05}^{0.4} Y_n^{\mathcal{E},u}(t_n + s)B(s)ds,$$

where $t_n = n \times 0.4 - 0.05$ is the starting time of the EEG segment $Y_n^{\mathcal{E},u}$. We denote by $Z_n^{\mathcal{E},B,u}$ the *"projection"* of the segment $Y_n^{\mathcal{E},u}$ in the *"direction"* $B$.

2. Denote $\mathcal{Z}_N^{\mathcal{E},B,u}$ the set of projections of the EEG segments belonging to $\mathcal{Y}_N^{\mathcal{E},u}$ in the direction $B$. To simplify the notation in the following we omit $\mathcal{E}$ unless otherwise noted.

3. Let $B_1, \ldots, B_M$ be a sequence of independent realizations of the Brownian Bridge indexed by the time interval $[-0.05, 0.4]$ and generated independently of the data set. For each pair of strings $u \in \mathcal{U}$ and $u' \in \mathcal{U}$, $u \neq u'$, we define the $\widehat{\Delta}_{M,N}^{\delta}(Q^u, Q^{u'})$ empirical dissimilarity as

$$\widehat{\Delta}_{M,N}^{\delta}(Q^u, Q^{u'}) = \frac{1}{M}\sum_{j=1}^{M}\mathbf{1}\{\|\widehat{F}_{u,B_j}^N - \widehat{F}_{u',B_j}^N\|_\infty > \delta\},$$

where

$$\widehat{F}_{u,B_j}^N(t) = \frac{1}{|\mathcal{Z}_N^{u,B_j}|}\sum_{z \in \mathcal{Z}_N^{u,B_j}}\mathbf{1}\{z \leq t\}$$

and

$$\widehat{F}_{u',B_j}^N(t) = \frac{1}{|\mathcal{Z}_N^{u',B_j}|}\sum_{z \in \mathcal{Z}_N^{u',B_j}}\mathbf{1}\{z \leq t\},$$

where $M$ is the number of random directions where the data is projected and $N$ is the length of the $X$ stimuli sequence.

This procedure is depicted in Fig 3.

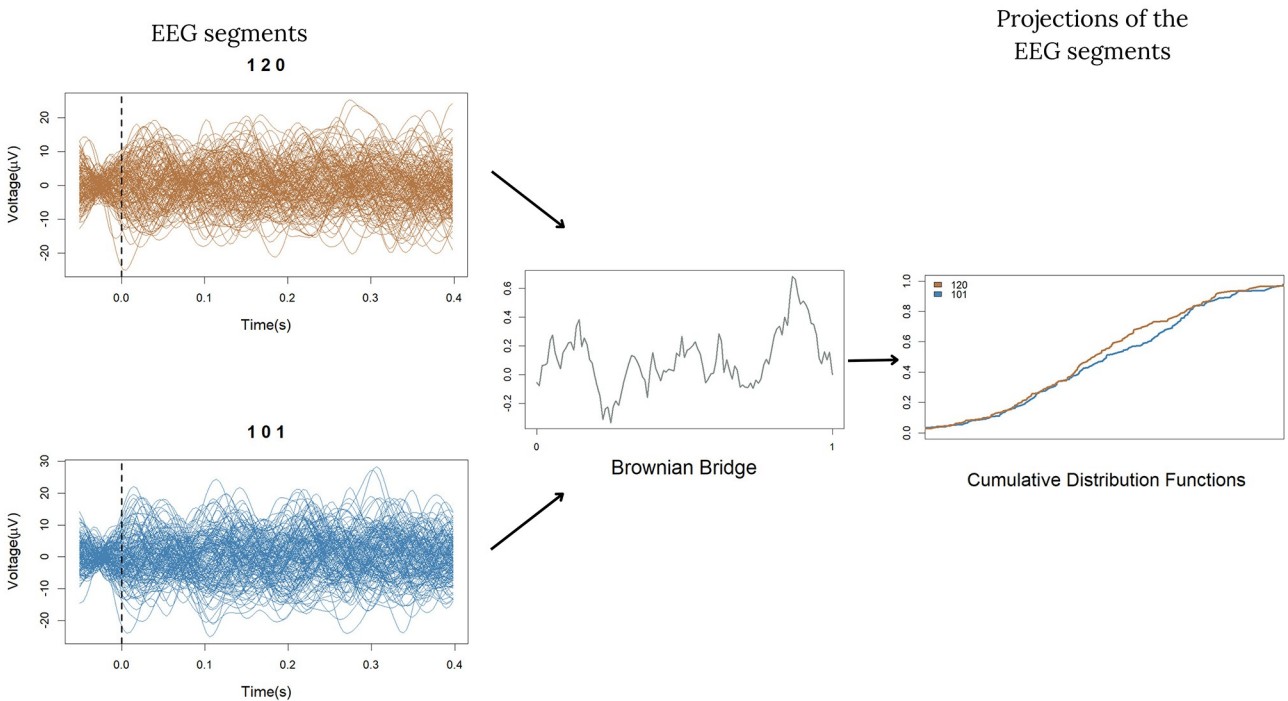

**Fig 3. Example of the projective method applied to sets of EEG segments (120 and 101) for a given participant and electrode.** All segments of both sets are projected into the same independent realization of a Brownian Bridge. For each set, we obtain a new set of real numbers. We can then compare the distribution of the two sets of real numbers obtained. Here we represent these distributions by their cumulative distribution functions and histograms.

For each participant $v$ and each subset $\mathcal{E} \subset E$, using the empirical dissimilarity matrix $\widehat{\Delta}^{\delta}_{M,N}$, we build a dendrogram of the set $\mathcal{U}$ using the sets $\{\mathcal{Y}^{u}_{N} : u \in \mathcal{U}\}$ of EEG segments by using the hierarchical clustering procedure with complete linkage (see Chapter 14, [26]). Complete linkage is an agglomerative clustering method. Initially, each element is a singleton, i.e. $u \in \mathcal{U}$. At each step, the closest clusters of sets of strings are merged. Given two clusters of sets of strings $C_1$ and $C_2$, which are subsets of $u \in \mathcal{U}$, the distance between them is

$$d(C_1, C_2) = \max_{u \in C_1, u' \in C_2} \widehat{\Delta}^{\delta}_{M,N}(Q^{u}, Q^{u'}).$$

The procedure ends when all sets of strings are put together into a single cluster. The dendrogram is a visual representation of this procedure in the form of a tree. At the bottom, each leaf represents an observation, and at the top, the entire data set is aggregated. The vertical axis shows the distance between two clusters at the step where they are merged.

For every pair of strings $(u, u')$ we compute the dissimilarity $\widehat{\Delta}^{\delta}_{M,N}(Q^{u}, Q^{u'})$ with $M = 5000$, as in [7]. We choose the threshold $\delta$ as the Kolmogorov-Smirnov statistic for level $\alpha = 0.05$ [27]. This equals to $\sqrt{\log(2/\alpha)/2|\mathcal{Z}^{u,B}_{N}|}$ whenever $|\mathcal{Z}^{u,B}_{N}| = |\mathcal{Z}^{u',B}_{N}|$. Since this is not always the case, we chose the threshold for each pair as the two-sided Kolmogorov Smirnov threshold described in [28] replacing $2|\mathcal{Z}^{u,B}_{N}|$ by $|\mathcal{Z}^{u,B}_{N}||\mathcal{Z}^{u',B}_{N}|/|\mathcal{Z}^{u,B}_{N}| + |\mathcal{Z}^{u',B}_{N}|$. Finally we obtain a partition of the sets $\{\mathcal{Y}^{u}_{N} : u \in \mathcal{U}\}$ of EEG segments using a fixed threshold $\gamma$. We select the coarser partition such that, for all pairs $(u, u')$ in each cluster, we have $\Delta^{\delta} < \gamma$. In our analysis we set $\gamma$ as the $v$-quantile of a binomial distribution with parameters $M$ and $\beta$. In our analysis we choose $\alpha = \beta = 1 - v = 0.05$, following the traditional significance level used in neurobiology. With these parameters we obtain $\gamma = 276/5000$. To test whether the significant results in our analysis depend on the choice of these parameters, we conducted new analyses replacing $\alpha$, $\beta$ and $v$ individually to 0.01.

**Summarizing partitions obtained from each participant.** For each fixed set of electrodes $\mathcal{E} \subset E$ the procedure we have just described produces a partition of the sets $\{\mathcal{Y}^{u} : u \in \mathcal{U}\}$ for each participant $v$. For each $v \in \mathcal{V}$, we denote as $\mathcal{C}^{v}$ the partition obtained with the procedure described above. Given two strings $u$ and $u'$, we say that $u \overset{v}{\sim} u'$, if the sets $\mathcal{Y}^{u}$ and $\mathcal{Y}^{u'}$ belong to the same cluster of the partition $\mathcal{C}^{v}$.

To summarise the results across participants we then perform an aggregation consensus clustering. By an aggregation consensus clustering we mean the following. We define a new dissimilarity matrix $\eta$ for the elements of $\mathcal{U}$ with entries defined as follows. For each pair $(u, u') \in \mathcal{U}^{2}$,

$$\eta(u, u') = \frac{1}{|\mathcal{V}|(|\mathcal{V} - 1|)} \sum_{(v,v') \in \mathcal{V}^{2}, v \neq v'} \mathbf{1}\{u \overset{v}{\sim} u'\}\mathbf{1}\{u \overset{v'}{\sim} u'\} .$$

This means that given two participants $v, v' \in \mathcal{V}$ and two strings $u$ and $u'$, we have that $\mathbf{1}\{u \overset{v}{\sim} u'\}\mathbf{1}\{u \overset{v'}{\sim} u'\} = 1$ if and only if the sets $\mathcal{Y}^{u}$ and $\mathcal{Y}^{u'}$ belong to the same cluster in both partitions $\mathcal{C}^{v}$ and $\mathcal{C}^{v'}$. Then $\eta(u, u')$ the average of participants for which $u$ and $u'$ belong to the same cluster.

Once we have the dissimilarity matrix $\eta$, we obtain a new dendrogram by using a hierarchical clustering procedure on the set $\mathcal{U}$ with the Ward linkage [29, 30]. We have 12 sets of strings to cluster. There are several criteria to automatically determine the number of clusters, usually based on comparing the internal cohesion of the clusters with the similarities between them [31]. In this case, we assume that there are at most five clusters to avoid multiple singletons.

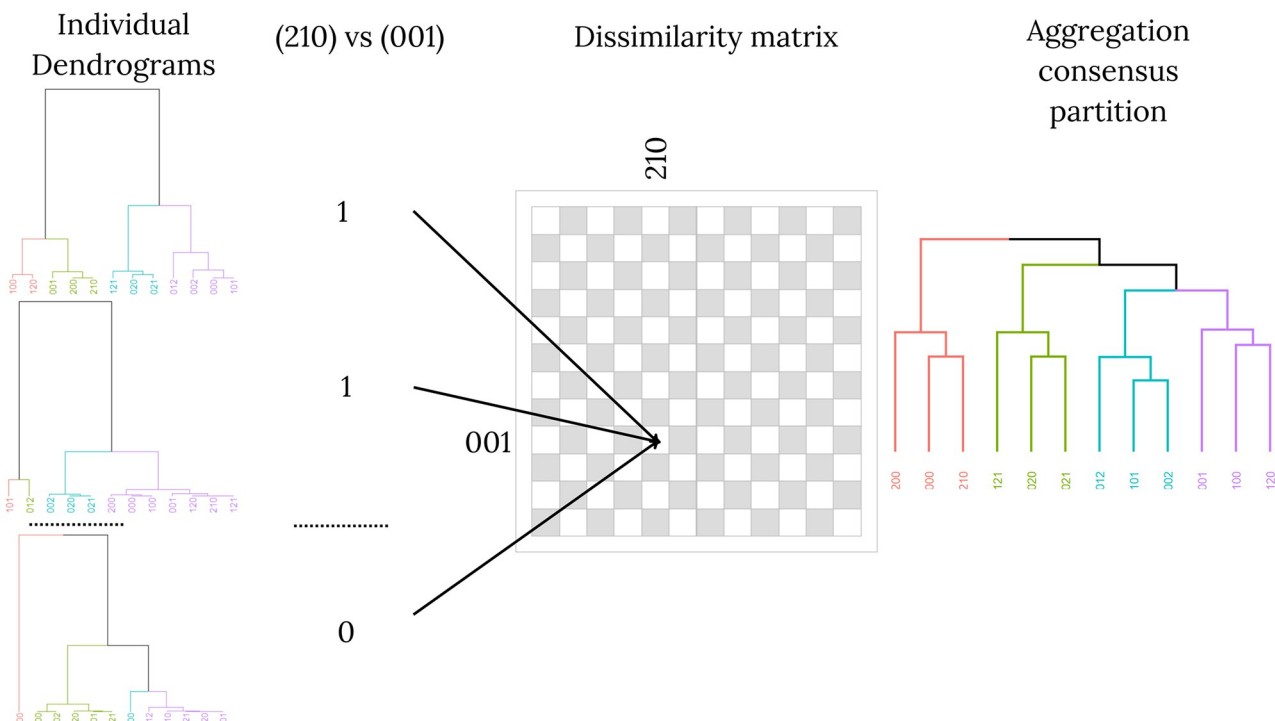

**Fig 4. Procedure for the construction of a consensus dendrogram.** For each individual dendrogram obtained with the method described in the section 'Clustering sets of EEG data' and any pair of strings in $\mathcal{U}$, we assign to each dendrogram 1 if the two strings are in the same cluster, and 0 otherwise. We take the average of these results as the dissimilarity between the two strings. Finally, using this dissimilarity matrix, we construct the consensus dendrogram.

We consider three well-known criteria to automatically determine the number of clusters: Calinsky-Harabaz (CH), Mean Silhouette (SI) and Dunn Index (DI) in our study. Therefore we investigate the consistency of the partitions obtained with three to five clusters. For the RPF region, CH selects 4 groups, SI finds a tie between 3 and 4 clusters and DI shows a tie between 4 and 5 groups. For the LPF region, CH selects 3 clusters, while DI shows a tie between 4 and 5 groups and SI results are almost tied between 3, 4 and 5 clusters. In the case of the control region OCC, both CH and DI select 5 clusters and SI shows very similar results for 3, 4 and 5 groups. We call the outputs of this procedure as *consensus dendrogram* and *consensus partition*. We denote a consensus partition by $P^{\mathcal{E}}$. This process is shown in Fig 4. All codes are accessible at https://github.com/fanajman/The-brain-uses-renewal-points-to-model-random-sequences-of-stimuli.git and the data is acessible at https://neuromat.numec.prp.usp.br/neuromatdb/EEGretrieving/.

**Statistical significance of the consensus partitions.** We conducted a numerical estimation of the probability of retrieving partitions under the following null hypothesis that the partitions were obtained purely at random and in a uniform way.

To compute numerically the p-values we generate $10^6$ random dissimilarity matrices in the following way. Let $R_i$: $1, \cdots, 10^6$ be random symmetric matrices in $\mathbb{R}^{12 \times 12}$, since in our case there are 12 strings $u = (u_1, u_2, u_3)$, where the entries of the upper diagonal of each matrix are independent uniform variables assuming values in the $[0, 1]$ interval. For each $R_i$, each entry in the symmetric matrix represent an independently generated dissimilarity between a pair $(u, u') \in \mathcal{U}^2$. For each random matrix $R_i$, we use it as the dissimilarity matrix between the 12

elements of $\mathcal{U}$ to construct a dendrogram using the Ward linkage [30] and select a partition $P_i^K$ with $K$ elements. In our main analysis we chose $K = 4$. The objective function used in Ward linkage is similar to k-means objective function, but the algorithms are different. This choice of linkage follows from the fact that using a central element of the clusters to determine the partition minimises the effect of outliers.

Given two partitions $P_1$ and $P_2$, we denote as $ARI(P_1, P_2)$ the value obtained by employing the adjusted Rand index [32], which is a measure of similarity between the between the pair $(P_1, P_2)$. For $P_1 = (P_{11}, \ldots, P_{1k})$ and $P_1 = (P_{21}, \ldots, P_{2k'})$ two partitions of the same non empty set with $q < \infty$ elements the ARI is defined as follows.

$$ARI(P_1, P_2) = \frac{\left( \sum_{(P_{1.}, P_{2.}) \in P_1 \times P2} c(|(P_{1.} \cap P_{2.}|)) \right) - \zeta(P_1, P_2)}{\frac{1}{2} \left( \sum_{P_{1.} \in P_1} c(|(P_{1.}|) + \sum_{P_{2.} \in P_2} c(|(P_{2.}|) \right) - \zeta(P_1, P_2)},$$

where for $j \in \mathbb{N}$, $c(j) = \frac{j!}{2!(j-2)!}$ and

$$\zeta(P_1, P_2) = \frac{\sum_{P_{1.} \in P_1} c(|P_{1.}|) \sum_{P_{2.} \in P_2} c(|P_{2.}|)}{c(q)}.$$

This statistic ranges in the $[-1, 1]$ interval, returning 1 for two identical partitions.

For each set of electrodes $\mathcal{E} \subset E$ we take the proportion

$$\widehat{p}(\mathcal{E}) = \frac{1}{10^6} \sum_{i=1}^{10^6} \mathbf{1}\{ARI(P_i, P^*) \geq ARI(P^*, P^{\mathcal{E}})\}$$

as the estimated probability of finding a partition at least as similar to the partition $P^*$ defined by the the occurrence of the strong beat, as described in subsection 'Clustering of EEG segments per participant'.

## Results

The consensus partitions $(P^{\mathcal{E}_{RPF}}, P^{\mathcal{E}_{LPF}}, P^{\mathcal{E}_{OCC}})$ with four clusters and their associated consensus dendrograms obtained from the sets of electrodes $(\mathcal{E}^{RPF}, \mathcal{E}^{LPF}, \mathcal{E}^{OCC})$ are shown in Fig 5.

Fig 5 left panel shows the consensus partition and associated consensus dendrogram retrieved from the subset $\mathcal{E}^{RPF}$. In this partition, 9 out of the 12 strings were classified as expected. The partition obtained contains the following clusters.

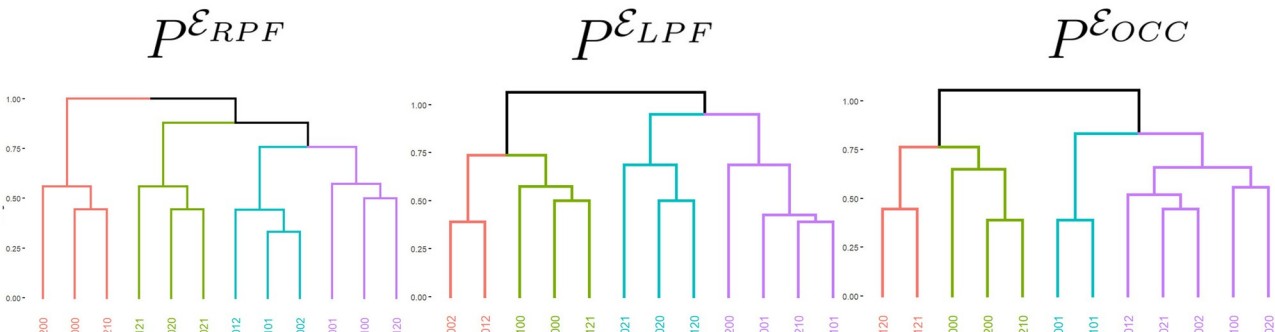

**Fig 5. Partitions and associated dendrograms retrieved from the EEG segments recorded in $\mathcal{E}^{RPF}$, $\mathcal{E}^{LPF}$ and $\mathcal{E}^{OCC}$.** In each dendrogram, the colours indicate the different clusters.

- A cluster with the strings 012 and 002, the two strings that end in the strong beat. This cluster also contains the string 101.

- A cluster with the strings 021, 020 and 121. These three strings end one position after the strong beat.

- A cluster with the strings 200 and 210. These are the two strings ending in a non random silent unit. This cluster also contains the string 000.

- A cluster with the strings 001 and 100. These strings end four positions after the strong beat. This cluster also contains the string 120.

Fig 5 central panel shows the consensus partition and associated consensus dendrogram retrieved from the set $\mathcal{E}^{LPF}$. This partition is similar to the right prefrontal electrodes in the sense that

- The two strings 002 and 012 ending in the strong beat are assigned to the same cluster.

- The two strings 200 and 210 ending in a non random silent unit are assigned to the same cluster.

- The consensus partition contains a cluster consisting of the strings 021, 020 and 120 ending one position after the strong beat.

Fig 5 right panel shows the partition and associated dendrogram obtained from the occipital electrodes, chosen as a control region. As in the prefrontal partitions, the two strings ending in the strong beat are assigned to the same cluster and the two strings ending in the non random silent unit are also assigned to the same cluster. The strings ending in the non random silent unit are assigned to a cluster which also contains two strings ending one step before the strong beat. No cluster containing exclusively strings ending one step after the strong beat were obtained.

A graphical representation of the concordance of the partitions $P^{\mathcal{E}_{LPF}}$ and $P^{\mathcal{E}_{RPF}}$ with respect to $P^*$ is given in Fig 6. Most sets of EEG strings with the strong beat in the same position remain in the same group as in $P^*$.

## Results on the statistical significance of the consensus partitions

For each partition retrieved from a subset of electrodes, the probability of finding a partition at least as similar as the partition expected was numerically estimated with the method described in the 'Statistical significance of the consensus partitions' section. The p-values are computed by $\widehat{p}(\mathcal{E})$. In italics are the p-values smaller than the usual 0.05 criterion used to consider the effect significant.

- Right prefrontal region: $\widehat{p}(\mathcal{E}^{RPF}) = $ *0.0174*.

- Left prefrontal region: $\widehat{p}(\mathcal{E}^{LPF}) = $ *0.0096*.

- Occipital region: $\widehat{p}(\mathcal{E}^{OCC}) = 0.1998$.

We also conducted the analysis for 3 and 5 clusters. The results in Table 1 reinforce the idea that the results obtained from the $\mathcal{E}^{RPF}$ and $\mathcal{E}^{LFP}$ agree with the partition $P^*$, whereas the $\mathcal{E}^{OCC}$ does not, regardless of the number of clusters.

To test whether the significance of the p-values obtained with these analyses depend on the $\alpha, \beta$ and $\nu$ parameters, we repeated the analysis three times, changing individually each parameter value from 0.05 to 0.01. Results are shown in Table 2.

## Pre-frontal Region

**Fig 6. Sankey diagram showing the matching between $P^{\mathcal{E}_{LPF}}$, $P^*$, and $P^{\mathcal{E}_{RPF}}$.** The middle column presents the $P^*$ partition. The colours represent the position of the strong beat in the string. On the left side of the diagram is the partition $P^{\mathcal{E}_{LPF}}$. Similarly, on the right-hand side is the partition corresponding to $P^{\mathcal{E}_{RPF}}$. The diagram is read from the centre outwards.

The p-values obtained with the set $\mathcal{E}^{RPF}$ are significant for the changes in $\beta$ to 0.01 and $v$ to 0.99 for all choices of cluster size. The p-values obtained with the set $\mathcal{E}^{LPF}$ stay significant when we change the value of $\alpha$ to 0.01 for four clusters and when we change $\beta$ to 0.01 for three clusters.

**Laterality and specificity of the results.** We employed two different analyses to investigate a possible lateralization component on the retrieved partitions from the prefrontal electrodes. In the first analysis we used the set of central prefrontal electrodes $\mathcal{E}^{CNT} = \{10, 11, 18\}$.

**Table 1. p-values for different numbers of clusters and regions analysed.**

| Number of clusters | RPF | LPF | OCC |
|---|---|---|---|
| 3 | *0.027* | *0.049* | 0.527 |
| 4 | *0.017* | *0.008* | 0.198 |
| 5 | *0.028* | *0.008* | 0.096 |

**Table 2. p-values for different parameters choices.**

| Region | Parameters | Number of clusters | | |
|---|---|---|---|---|
| | | 3 | 4 | 5 |
| RPF | $\alpha = \mathbf{0.01}, \beta = 0.05, \nu = 0.95$ | 0.239 | 0.204 | 0.266 |
| | $\alpha = 0.05, \beta = \mathbf{0.01}, \nu = 0.95$ | *0.001* | *0.002* | *0.003* |
| | $\alpha = 0.05, \beta = 0.05, \nu = \mathbf{0.99}$ | *0.025* | *0.018* | *0.026* |
| LPF | $\alpha = \mathbf{0.01}, \beta = 0.05, \nu = 0.95$ | 0.080 | *0.021* | 0.304 |
| | $\alpha = 0.05, \beta = \mathbf{0.01}, \nu = 0.95$ | *0.019* | 0.208 | 0.067 |
| | $\alpha = 0.05, \beta = 0.05, \nu = \mathbf{0.99}$ | 0.195 | 0.081 | 0.097 |

The position of the electrodes is shown in Fig 1. The dendrogram of the retrieved partition appears in Fig 7 left panel.

The consensus partition obtained in the analysis corresponding to $\mathcal{E}^{CNT}$ is close to the partition $P^*$ obtained using the position of the strong beat. We can observe the following similarities.

- There is a cluster with strings ending in the strong beat, 012 and 002. This cluster also contains the string 001.

- There is a cluster with the strong beat in the middle position containing the strings 120, 020 and 021.

- There is a cluster with strings 200 and 210. This cluster also contains the string 101.

- The remaining strings, 121, 100 and 000, are clustered together.

The second strategy consisted of removing electrode 11 from the analysis using instead the two sets of two electrodes $\mathcal{E}^{LRPF} = \{9, 10\}$ and $\mathcal{E}^{LLPF} = \{18, 22\}$. The corresponding dendrograms are depicted in Fig 7 centre and right panels respectively.

We observe some important similarities between the partitions obtained with the sets $\mathcal{E}^{LRPF}$ and $\mathcal{E}^{LLPF}$ and $P^*$.

- In both cases the strings ending in two, 002 and 012, are in the same cluster.

- The strings 021 and 121 are assigned to the same cluster. Both strings end in a weak beat preceded by a strong beat.

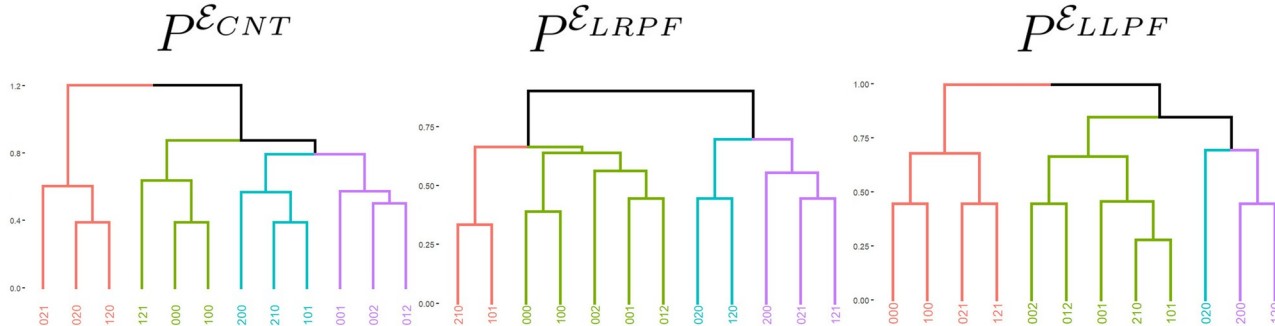

**Fig 7. Partitions and associated dendrograms retrieved from the EEG segments recorded in $\mathcal{E}^{CNT}$, $\mathcal{E}^{LRPF}$ and $\mathcal{E}^{LLPF}$.** In each dendrogram, the colours indicate the different clusters.

**Table 3. p-values for different numbers of clusters and regions analysed.**

| Number of clusters | CNT | LRPF | LLPF |
|---|---|---|---|
| 3 | 0.065 | *0.034* | 0.279 |
| 4 | *0.018* | *0.042* | 0.312 |
| 5 | *0.007* | 0.064 | 0.132 |

- In the LRPF we also have the strings 020 and 120 assigned to the same cluster.

- Also in the LRPF the three strings 000, 100 and 001 are assigned to the same cluster. All these strings end one step before a strong beat.

To determine if the consensus partitions found are statistically similar to the four cluster partition $P^*$ expected by the occurrence of the strong beat, we tested the null hypothesis that the results obtained are by chance for 3, 4 and 5 clusters. Table 3 exhibits these estimated p-values.

For the CNT set of electrodes we obtain a p-value smaller than 0.05 for 4 and 5 classes. In the LRPF partition the p-values are all close to 0.06, with 3 and 4 clustering structures resulting in p-values below the 0.05 threshold. However, for LLFP we obtain large p-values, all greater than 0.13 which indicates that the left leaning electrodes do not capture the $P^*$ structure at a significant level. Remarkably, when considering a 4 cluster structure a statistically significant result is attained for both CNT and LRPF.

## Discussion

The consensus partitions retrieved from prefrontal EEG data show significant similarity to the partition expected by the occurrence of the strong beat. These results agree with the conjecture that the brain uses the strong beat represented by the symbol 2 to model the stimuli sequence. More precisely, the brain seems to select a partition of the past of the stimuli sequence with four clusters, this partition being defined by the last occurrence of the strong beat.

The use of the occurrence of the strong beat to model the stimuli sequence indicates that the brain is able to find a model of the sequence, that is, to identify the structural aspects of the stimuli sequence to accurately estimate the next event amidst the stochasticity. A partition of the past of the sequence of stimuli with only four elements is an economical representation of the sequence, since this is the smallest partition of the past which allows generating the stimuli sequence step-by-step.

As in [7], we found that the information of temporal dependencies of the stochastic sequence were encoded in the electrodes positioned over the prefrontal cortex. The results obtained for $\mathcal{E}^{RPF}$ and $\mathcal{E}^{LPF}$ are significantly similar to the expected partition $P^*$. Moreover, when removing the central electrode 11 from the analysis we obtained a partition structure similar to $P^*$ only for the right leaning set of electrodes $\mathcal{E}^{LRPF}$ indicating a slight lateralization. For the occipital electrodes we did not find a correspondence between the clusters obtained and the partition $P^*$ determined by the position of the strong beat. This is expected, since this cortical region is mostly associated with visual processing [23]. It is interesting to note that the strings ending in a strong beat were assigned to the same cluster in the partition obtained for $\mathcal{E}^{OCC}$.

In [7], a context tree model approach was used to identify the structure of the sequence of auditory stimuli encoded in the EEG data. In comparison, employing a new clustering procedure for sets of functional data, we tested for all pairs of data sets indexed by the strings in $\mathcal{U}$ which data sets had the same law. Our results agree with a partition that has four elements, in

contrast with the eight elements of the context tree model proposed in [7], indicating that our approach can identify this more efficient compression scheme.

We also tested the sensibility of this method for our choices of parameters. The results obtained with the set $\mathcal{E}^{RPF}$ were still significant for the more conservative choices of $\beta = 0.01$ and $1 - \nu = 0.01$. While the results obtained with the set $\mathcal{E}^{LPF}$ showed less agreement with the expected partition, we note that if we use a less strict 0.1 as our threshold for the final significance in Table 2, we obtain significant results for most combinations of number of clusters and choice of parameters. The loss of signal obtained for $\alpha = 0.01$ can be a consequence of the Kolmogorov Smirnov test being somewhat conservative for multiple comparisons [33]. Even though these tests show that our result is consistent, whether the choice of Gaussian measure $P$ affects the results is a topic that could lead to interesting new studies.

The occurrence of the strong beat regenerates the sequence, meaning that the stimuli sequence can be partitioned in independent blocks demarcated by the occurrence of each strong beat. It is a well known fact that participants can perceive sequences of stimuli as independent blocks, a phenomenon called chunking (for a review see [34]). In music the usual rhythm patterns have a strong beat that presents the regenerative property by occurring periodically [35]. Therefore our conjecture gives us a natural reason for this phenomenon. However, the fact that the strong beat is at the same time periodic and regenerative precludes to state which of these two complementary properties are determinant for obtaining the partition.

In [36] it is conjectured that the activity of some neuron populations synchronizes with temporally regular occurrences in the auditory stimuli, a phenomenon sometimes called entrainment. This is a possible mechanism used by the brain to model the stimuli sequence, considering that in our experiment the isochronous occurrence of a strong beat every four auditory units is a recurrent regularity. The entrainment phenomenon has been shown to explain chunking of continuous auditory stimuli in the temporal scale of 150–300 ms in the temporal cortex [37]. Our results indicate that the prefrontal cortex uses the occurrence of the strong beat to chunk the auditory sequence in larger timescales, with seconds between the occurrence of each strong beat.

We have introduced a new clustering procedure that enables grouping sets of EEG data by their law. This was done employing the dissimilarity measure $\Delta^{\delta}$, which is based on the Kolmogorov-Smirnov distance. Other dissimilarities between probability measures, such as Wasserstein's distance or Kullback Leiber's divergence could be considered. For both cases there are goodness-of-fit test proposals for multivariate data [38, 39], however how to adapt these measures to cluster functional data sets is unclear since, to the best of our knowledge, there are no proposals able to deal with data such as square-integrable functions.

## Conclusion

We show evidence that the brain uses the occurrence of strong beat of a stochastic sequence of auditory stimuli to select a model of a random sequence of stimuli.

## Acknowledgments

The authors acknowledge the hospitality of the Institut Henri Poincaré (LabEx CARMIN ANR-10-LABX-59-01) where part of this work was written and Jorge Stolfi for his insightful comments on the manuscript.

**Dedicated to the memory of Antonio Galves**.

## Author Contributions

**Conceptualization:** Fernando A. Najman, Antonio Galves, Claudia D. Vargas.

**Data curation:** Fernando A. Najman, Antonio Galves, Marcela Svarc, Claudia D. Vargas.

**Formal analysis:** Fernando A. Najman, Antonio Galves, Marcela Svarc.

**Funding acquisition:** Antonio Galves, Claudia D. Vargas.

**Investigation:** Fernando A. Najman, Antonio Galves, Marcela Svarc, Claudia D. Vargas.

**Methodology:** Fernando A. Najman, Antonio Galves, Marcela Svarc.

**Project administration:** Fernando A. Najman, Antonio Galves, Marcela Svarc, Claudia D. Vargas.

**Resources:** Antonio Galves, Claudia D. Vargas.

**Software:** Fernando A. Najman.

**Supervision:** Antonio Galves, Claudia D. Vargas.

**Validation:** Fernando A. Najman, Antonio Galves, Marcela Svarc, Claudia D. Vargas.

**Visualization:** Fernando A. Najman, Antonio Galves, Marcela Svarc, Claudia D. Vargas.

**Writing – original draft:** Fernando A. Najman, Antonio Galves, Marcela Svarc, Claudia D. Vargas.

**Writing – review & editing:** Fernando A. Najman, Antonio Galves, Marcela Svarc, Claudia D. Vargas.

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
