## [Decision Letter · Decision Letter 0]

8 May 2024

Dear Professor Vargas,

Thank you very much for submitting your manuscript "The brain uses renewal points to model random sequences of stimuli." for consideration at PLOS Computational Biology.

As with all papers reviewed by the journal, your manuscript was reviewed by members of the editorial board and by several independent reviewers. In light of the reviews (below this email), we would like to invite the resubmission of a significantly-revised version that takes into account the reviewers' comments.

The paper is devoted to verify on output EEG data that they are consistent with a certain stochastic model of the input signal,

that fits the actual data generation mechanism. This subject of research has undoubtedly a large interest, as testified by the reports of all the referees. Indeed, some of the authors of the paper under consideration have already published a previous paper (number 5 in their reference list), where a more complicate model of the input is assumed, which is still compatible with the generation of the data. Of course various different statistical models can be validated on the same data, but several criteria to choose among them have been proposed. As referee 2 is asking, the authors should compare the results of the two papers. On the basis of this remark, too assertive statements about the conclusions of an experimental work of this kind should be presented with caution, as Referee 1 suggests.

Both referees 2 and 3 are asking for a clearer explanation, with the necessary biological material, of the "partition" of the location of the electrodes in RPF, LPF and OCC (from the mathematical point of view this is not a partition since 11 belongs both to RPF and LPF).

But a good wealth of doubts by all referees concerns the clustering procedure used in the analysis of the experimental data. First of all the dissimilarity measure suggested by the authors depend on the parameter delta, whose choice seems to fall somewhat from the sky. In addition, the definition of Delta itself seems unclear. What is P? It selects a random b is L^2[0.T] (I suspect here [0,T] is [-0.05, 0.4], correct?), but then this b should be a common parameter of the compared CDF's, but which? Looking at its estimator does not help, since one of the parameter it depends on an n which does not even appear in the definition (see the remarks of referee 2).

Concerning the clustering procedure itself, some further discussion seems needed (referee 2).

The question raised by Referee 3 about the aggregation procedure is very natural: isn't it that eta is the square of the number of v's with u and u' in the same cluster? And, more importantly, how the final partition is selected from the consensus dendrogram? It seems that the number of cluster is externally constrained to be 4, which somewhat weakens the conclusion of the analysis. As the last point raised by Referee 1 indicates, it seems to me that we end up with a partition of input strings, not of output chunks.

A serious point raised both by referee 2 and 3 is that the renewal point of the input sequence is happening periodically, which makes impossible to discriminate, on the basis of this experimental protocol, if the results obtained are due to the regeneration or to the periodicity. The Authors should say something on this delicate issue.

Summarizing all the previous points, I believe that the paper needs a major revision to be publishable. Even if I have underlined only those more significant to me, I urge The Authors to care about all of them.

We cannot make any decision about publication until we have seen the revised manuscript and your response to the reviewers' comments. Your revised manuscript is also likely to be sent to reviewers for further evaluation.

Sincerely,

Mauro Piccioni

Guest Editor

PLOS Computational Biology

Lyle Graham

Section Editor

PLOS Computational Biology

The paper is devoted to verify on output EEG data that they are consistent with a certain stochastic model of the input signal,

that fits the actual data generation mechanism. This subject of research has undoubtedly a large interest, as testified by the reports of all the referees. Indeed, some of the authors of the paper under consideration have already published a previous paper (number 5 in their reference list), where a more complicate model of the input is assumed, which is still compatible with the generation of the data. Of course various different statistical models can be validated on the same data, but several criteria to choose among them have been proposed. As referee 2 is asking, the authors should compare the results of the two papers. On the basis of this remark, too assertive statements about the conclusions of an experimental work of this kind should be presented with caution, as Referee 1 suggests.

Both referees 2 and 3 are asking for a clearer explanation, with the necessary biological material, of the "partition" of the location of the electrodes in RPF, LPF and OCC (from the mathematical point of view this is not a partition since 11 belongs both to RPF and LPF).

But a good wealth of doubts by all referees concerns the clustering procedure used in the analysis of the experimental data. First of all the dissimilarity measure suggested by the authors depend on the parameter delta, whose choice seems to fall somewhat from the sky. In addition, the definition of Delta itself seems unclear. What is P? It selects a random b is L^2[0.T] (I suspect here [0,T] is [-0.05, 0.4], correct?), but then this b should be a common parameter of the compared CDF's, but which? Looking at its estimator does not help, since one of the parameter it depends on an n which does not even appear in the definition (see the remarks of referee 2).

Concerning the clustering procedure itself, some further discussion seems needed (referee 2).

The question raised by Referee 3 about the aggregation procedure is very natural: isn't it that eta is the square of the number of v's with u and u' in the same cluster? And, more importantly, how the final partition is selected from the consensus dendrogram? It seems that the number of cluster is externally constrained to be 4, which somewhat weakens the conclusion of the analysis. As the last point raised by Referee 1 indicates, it seems to me that we end up with a partition of input strings, not of output chunks.

A serious point raised both by referee 2 and 3 is that the renewal point of the input sequence is happening periodically, which makes impossible to discriminate, on the basis of this experimental protocol, if the results obtained are due to the regeneration or to the periodicity. The Authors should say something on this delicate issue.

Summarizing all the previous points, I believe that the paper needs a major revision to be publishable. Even if I have underlined only those more significant to me, I urge The Authors to care about all of them.

Reviewer's Responses to Questions

**Comments to the Authors:**

Reviewer #1: The methodology presented for analyzing the EEG data is novel and holds great potential.

Mayor point

Please make a revision on the following point:

- From my standpoint, the title seems overly assertive and lacks complete validation based on the presented results.

The same when the authors say: "This clustering procedure indicates that the brain uses renewal points in the stochastic sequence of auditory stimuli in order to build a model."

"The clusters obtained from the experimental data show that the strong beat of the rhythmic structure is used by the brain to encode the sequence." And similar sentences in the manuscript.

I think that it is really very difficult to prove what is stated.

I think the results could be interpreted in terms of the strength of the beat rather than being directly linked to higher-order encoding due to the stimulus. I believe more control experiments are necessary to support the authors' assertions, which is why I suggest toning down the interpretation.

Minor points:

Section Clustering sets of EEG segments

- virtual electrode could be replaced with arbitrary electrode.

- Figures 3, 4 and 6 could be combined into a single figure with three panels.

- Figure 5 could be included as an auxiliary figure in the Supplementary Material.

- The section finishes by saying:: "Using a fixed threshold we finally obtain a partition of the sets {Yu : u ∈ U} of EEG segments, which we denote by C." Please clarify.

Do you mean? A threshold is selected in the dendrogram for each condition to create a partition with four clusters. These partitions are denoted by C.

Reviewer #2: Please find my comments in the attached document.

Reviewer #3: The article is very interesting but lack many explanations. Let me list them below

- Page 2, Introduction, could you also compare your work to the work of Friston to understand how far it is from your present analysis ?

- page 3, I think there is a problem line 50, I think it is a 0 (see the sequence below)?

- Page 3, on preprocessing, did you also remove ocular artefacts ?

- Page 3, more importantly, how did you select the set of 8 electrodes ??? what would you find with another set ? Also explain why you focus on RPF LPF and OCC and not other Regions of Interest. By the way, say in full letters what these regions are the first time we see it in the paper.

- Page 4, explain what are $\\pi_B(Q)$ and $P(b)$. Say that $\\Delta^\\delta(Q,Q')$ is not computable in practice. Explain that you are using an estimate of it.

- page 5, the choice of $\\delta$ and the threshold where you cut your dendrogram seem to be superimportant for the building of the individual dendrograms. How did you choose these ? Not just the values, but why these values ? Also do we agree that at this stage, you might have more (or less) than 4 clusters ?

- page 5, If I understand qualitatively the grouping, why did you choose $\\eta(u,u')$ ? and not more basically, the number of individuals with u and u' in the same partition of their individual dendrograms ?

- page 5 again, the end, why forcing 4 clusters ? I understand that you want to compare to P* for you assumption but it might happen that the brain uses less or more, no ? a rule of the thumb might say that we cut the dendogram at the largest jump...

- page 6 and 7, why do you think the RPF partition is different than the LPF partition, is it just some random fluctuations ? is there a biological explanation ?

- Also there is no comment on the fact that nothing is detected in OCC ? could you explain it from a bioloical point of view ?

- I find the last paragraph a bit contradictory with the title. The biological explanation might be that, as you said, the brain follows the regular occurrences, (here the 2)... It just happens that in this case the 2 is also the renewal point of the random sequence, but we could totally imagine a sequence with a cyclic pattern and renewal points that would not coincide. So maybe this experiment is not enough to conclude with respect to renewal point. Or maybe I'm missing some more explanation here ?

**Have the authors made all data and (if applicable) computational code underlying the findings in their manuscript fully available?**

Reviewer #1: Yes

Reviewer #2: Yes

Reviewer #3: Yes

PLOS authors have the option to publish the peer review history of their article (what does this mean?). If published, this will include your full peer review and any attached files.

Reviewer #1: No

Reviewer #2: No

Reviewer #3: No
---

## [Decision Letter · Decision Letter 1]

7 Aug 2024

Dear Vargas,

Thank you very much for submitting your manuscript "The brain uses recurrent occurrences to model random sequences of auditory stimuli." for consideration at PLOS Computational Biology.

As with all papers reviewed by the journal, your manuscript was reviewed by members of the editorial board and by several independent reviewers. In light of the reviews (below this email), we would like to invite the resubmission of a significantly-revised version that takes into account the reviewers' comments.

We cannot make any decision about publication until we have seen the revised manuscript and your response to the reviewers' comments. Your revised manuscript is also likely to be sent to reviewers for further evaluation.

Sincerely,

Mauro Piccioni

Guest Editor

PLOS Computational Biology

Lyle Graham

Section Editor

PLOS Computational Biology

Reviewer's Responses to Questions

**Comments to the Authors:**

Reviewer #1: The authors have addressed all my questions and remarks. I now recommend

the publication of the new manuscript in Plos Computational Biology

Reviewer #2: Please find my attached comments.

Reviewer #3: Thank you for taking care of my comments. I'm mostly pleased with the changes but I have still several problems with the manuscript as it is

- The main problem is that if I prefer that you do not really refer to renewal points, I have doubts about what you mean by "recurrent occurences". Could you define it more precisely somewhere and why is 2, the recurrent occurence that match your definition on the sequence (and the only one)?

- I think your eta(u,u') is not the number of v for which u~u' but this number to the square. Again why don't you use the number directly ?

- There are at least 3 steps in your procedure

-> you choose a delta

-> you choose another threshold for the average over the realisations of the brownian bridge ( line 147 ). Both of these gives you a dendogram / partition per subject

-> you have another procedure to aggregate that over subject and find another partition for the whole population

I f I understand that you checked the sensibility for the third aspect. I don't see anywhere a discussion on the influence of the first two points. The delta seems to be taken due to KS distribution but if you change the 0.05 in line 142 -143 to 0.01 or whatever, what does it change ?

For the other threshold, it does not even have a name, whereas it seems to be super important for the sequel as well.

**Have the authors made all data and (if applicable) computational code underlying the findings in their manuscript fully available?**

Reviewer #1: **No: **

Reviewer #2: **No: **Neither the code nor the data have been shared.

Reviewer #3: None

PLOS authors have the option to publish the peer review history of their article (what does this mean?). If published, this will include your full peer review and any attached files.

Reviewer #1: No

Reviewer #2: No

Reviewer #3: No
---

## [Decision Letter · Decision Letter 2]

10 Dec 2024

PCOMPBIOL-D-23-01900R2

Extracting the fingerprints of sequences of random rhythmic auditory stimuli from electrophysiological data.

PLOS Computational Biology

Dear Dr. Vargas,

Thank you for submitting your manuscript to PLOS Computational Biology. After careful consideration, we feel that it has merit but does not fully meet PLOS Computational Biology's publication criteria as it currently stands. Therefore, we invite you to submit a revised version of the manuscript that addresses the points raised during the review process.

Please submit your revised manuscript within 30 days Feb 09 2025 11:59PM. If you will need more time than this to complete your revisions, please reply to this message or contact the journal office at ploscompbiol@plos.org. Please include the following items when submitting your revised manuscript:

We look forward to receiving your revised manuscript.

Kind regards,

Mauro Piccioni

Guest Editor

PLOS Computational Biology

Lyle Graham

Section Editor

PLOS Computational Biology

Feilim Mac Gabhann

Editor-in-Chief

PLOS Computational Biology

Jason Papin

Editor-in-Chief

PLOS Computational Biology

**Journal Requirements:**

1) Please amend your detailed Financial Disclosure statement. This is published with the article. It must therefore be completed in full sentences and contain the exact wording you wish to be published.

1) State the initials, alongside each funding source, of each author to receive each grant. For example: "This work was supported by the National Institutes of Health (####### to AM; ###### to CJ) and the National Science Foundation (###### to AM).".

**Comments to the Authors:**

**Please note that the reviews are uploaded as attachments.**

**Reviewers' comments:**

Reviewer's Responses to Questions

Reviewer #2: Please find attached my detailed report.

Reviewer #3: I'm mostly happy but there is still a problem in the definition of eta. I wrote it below as latex formulas

<math xmlns="http://www.w3.org/1998/Math/MathML"><semantics><mi>η</mi><mo>(</mo><mi>u</mi><mo>,</mo><mi>u</mi><mo>'</mo><mo>)</mo><mo>=</mo><munder><mo>∑</mo><mrow><mo>(</mo><mi>v</mi><mo>,</mo><mi>v</mi><mo>'</mo><mo>)</mo><mo>∈</mo><msup><mi>V</mi><mn>2</mn></msup></mrow></munder><msub><mn>1</mn><mrow><mi>u</mi><msup><mo> </mo><mi>v</mi></msup><mi>u</mi><mo>'</mo></mrow></msub><msub><mn>1</mn><mrow><mi>u</mi><msup><mo> </mo><mi>v</mi></msup><mo>'</mo><mi>u</mi><mo>'</mo></mrow></msub><mo>=</mo><munder><mo>∑</mo><mrow><mi>v</mi><mo>∈</mo><mi>V</mi></mrow></munder><msub><mn>1</mn><mrow><mi>u</mi><msup><mo> </mo><mi>v</mi></msup><mi>u</mi><mo>'</mo></mrow></msub><munder><mo>∑</mo><mrow><mi>v</mi><mo>'</mo><mo>∈</mo><mi>V</mi></mrow></munder><msub><mn>1</mn><mrow><mi>u</mi><msup><mo> </mo><mi>v</mi></msup><mo>'</mo><mi>u</mi><mo>'</mo></mrow></msub><mo>=</mo><mo>(</mo><munder><mo>∑</mo><mrow><mi>v</mi><mo>∈</mo><mi>V</mi></mrow></munder><msub><mn>1</mn><mrow><mi>u</mi><msup><mo> </mo><mi>v</mi></msup><mi>u</mi><mo>'</mo></mrow></msub><msup><mo>)</mo><mn>2</mn></msup><mo>.</mo><annotation encoding="LaTeX"> \\eta(u,u')=\\sum_{(v,v')\\in V^2} 1_{u~^v u'} 1_{u~^v' u'}= \\sum_{v\\in V} 1_{u~^v u'} \\sum_{v'\\in V} 1_{u~^v' u'}= (\\sum_{v\\in V} 1_{u~^v u'})^2.</annotation></semantics></math>

So it is a number to the square and not the number

**Have the authors made all data and (if applicable) computational code underlying the findings in their manuscript fully available?**

Reviewer #2: **No: **No computational code is provided

Reviewer #3: Yes

PLOS authors have the option to publish the peer review history of their article (what does this mean?). If published, this will include your full peer review and any attached files.

Reviewer #2: No

Reviewer #3: No

**Figure resubmission:**
---

## [Editor Report · Decision Letter 3]

2 Jan 2025

Dear Dr Vargas,

We are pleased to inform you that your manuscript 'Extracting the fingerprints of sequences of random rhythmic auditory stimuli from electrophysiological data.' has been provisionally accepted for publication in PLOS Computational Biology.

Best regards,

Mauro Piccioni

Guest Editor

PLOS Computational Biology

Lyle Graham

Section Editor

PLOS Computational Biology

Row 159, page 5. Errata: where is drawn Corrige: with b drawn

---

## [Editor Report · Acceptance letter]

7 Jan 2025

PCOMPBIOL-D-23-01900R3 

Extracting the fingerprints of sequences of random rhythmic auditory stimuli from electrophysiological data.

Dear Dr Vargas,

I am pleased to inform you that your manuscript has been formally accepted for publication in PLOS Computational Biology. Your manuscript is now with our production department and you will be notified of the publication date in due course.

With kind regards,

Anita Estes
